

# Groundwater level changes on Jeju Island associated with the Kumamoto and Gyeongju earthquakes

Soo-Hyoung Lee[1], Yoon-Suk Park[2], Kyoochul Ha[1], YongCheol Kim[1], Sung-Wook Kim[3], Se-Yeong Hamm[4*]

[1]Korea Institute of Geoscience and Mineral Resources, 124 Gwahak-ro, Yuseong-gu, Daejeon 34132, South Korea
[2]Jeju Special Self-Governing Province, 30 Munnyeon-ro, Jeju 63121, South Korea
[3]Geo-Information Institute, 1048-11 Jungang-daero, Yeonje-gu, Busan 47598, South Korea
[4]Department of Geological Sciences, Pusan National University, Busan 46241, South Korea

*Correspondence to*: Se-Yeong Hamm (hsy@pusan.ac.kr)

**Abstract.** The largest earthquake since the beginning of instrumental earthquake monitoring (magnitude 5.4) in Korean peninsula occurred in Gyeongju City area, South Korea, at 20:32:54 on September 12, 2016 (local time). Before the Gyeongju earthquake, an earthquake of magnitude 7.0 occurred in Kumamoto prefecture, Kyushu, Japan, at 01:25:06 on April 16, 2016 (local time). This study examined groundwater level changes of the monitoring wells on Jeju Island in relation to the Gyeongju and Kumamoto earthquakes. Groundwater level changes due to the Kumamoto and Gyeongju earthquakes exhibited spikes or oscillations, with the initial water level change occurring 2-3 min after earthquake generation, displaying different behaviors depending on the magnitude of the earthquakes and different sensitivities depending on the aquifer and geological characteristics. On Jeju Island, the groundwater level change caused by the Gyeongju earthquake (M 5.4) was larger than that caused by the Kumamoto earthquake (M 5.4). This was because a smaller energy attenuation occurred during the Gyeongju earthquake along the Yangsan fault on the Korean peninsula extending in the NNE-SSW direction, while a larger energy attenuation occurred during the Kumamoto earthquake along the median tectonic line (MTL) fault on the Japanese island arc extending in the ENE-WSW direction.

**1 Introduction**

A magnitude (M) 5.4 earthquake that occurred in Gyeongju, South Korea at 20:32:54, on September 12, 2016 (local time) was recorded as the largest earthquake in South Korea since the beginning of instrumental earthquake observation in 1978 (Kim et al., 2016). Over 500 aftershocks occurred and some are still underway (Fig. 1). The Gyeongju earthquake took place inside the Yangsan fault zone, extending in the NNE-SSW

direction. The Yangsan fault is one of the well-known dextral strike-slip faults in Korean peninsula, with a width of a few hundred meters to 2 km and a length of ~200 km from Yeonghae in the north to the western part of Busan City in the south; it also has an extremely well-developed topographic expression.

Earthquakes generate static stress and dynamic stress (or seismic waves) (Lay and Wallace, 1995). Numerous seismologists and hydrogeologists have verified that the stresses caused by earthquake are associated with

hydrological and hydrogeological changes, such as liquefaction of sediments, groundwater level change, the





increase of stream discharge, chemical composition change in groundwater, and the generation of new springs and mud volcanoes (Blanchard and Byerly, 1935; Wakita, 1975; Rojstaczer et al., 1995; Muir-Wood and King, 1993; Quility and Roeloffs, 1997). Stress changes in rocks caused by an earthquake produce strains that cause fluid pressure changes, which again change the hydrogeological properties such as groundwater level and

hydraulic conductivity (Manga and Wang, 2007). Earthquakes produce hydrogeological changes: physical changes including fluid pressure change (King et al., 1999; Chia et al., 2001; Jonsson et al., 2003; Matsumoto et al., 2003; Roeloffs et al., 2003; Wang and Chia, 2008) and geochemical changes (Ma et al., 1990; Woith et al., 2003; Charmoille et al., 2005; Wang et al., 2012). Seismic waves eliminate micro-cracks and micro-particles in fracture zones and generate chemical changes in initial groundwater by the inflow of new groundwater, the

acceleration of water-rock reaction, and fluid-source switching (Claesson et al., 2004, 2007; Skelton et al., 2014).

In addition, the dynamic strain caused by an earthquake (or seismic wave) affects the groundwater level several thousands kilometers away, and the groundwater level shows a fluctuation pattern similar to the seismic wave along the compression and expansion of the seismic wave (Cooper et al., 1965; Liu et al., 1989; Brodsky et al., 2003; Wang et al., 2009). For instance, groundwater level changes induced by the 2004 Sumatra

earthquake in Indonesia were observed in Fairbanks, Alaska, about 10,000 km from the source (Sil and Freymueller, 2006).

This study examined the changes in groundwater level on Jeju Volcanic Island caused by the Kumamoto earthquake that occurred in Japan at 01:25:06, on April 16, 2016 (local time) and the M 5.4 Gyeongju earthquake that occurred in Korea at 19:44:32, on September 12, 2016 (local time).

**2 Hydrogeology and Monitoring Wells**

Jeju Island, about 140 km south of the Korean peninsula, with a total area of 1,828 km2, is geographically located west of Japan. The island is a dormant shield volcano with one central mountain peak, Mt. Halla, rising to an elevation of 1950 m (Won et al., 2006). The stratigraphic units of the island are categorized into basement rocks (granite and welded tuff), U-formations (unconsolidated sediments), Seogwipo formations (conglomeratic

sandstone, sandstone, sandy mudstone, and mudstone with abundant bio-clastic shells), and basaltic and trachytic lavas (Yoon, 1999). Hydraulic parameters of the geological formations on Jeju Island have been estimated by using specific capacities from time-drawdown data (Hamm et al., 2005) and tidal response (Kim et al., 2005).

The Yangsan fault is one of the well-known dextral strike-slip faults in Korea and has an extremely well-

developed topographic expression, with a length of about 200 km between Yanghae in the north and western Pusan in the south and a width of a few hundred meters to 2 km. The strike of the fault zone generally extends in the direction of NNE in the south and N-S in the north, with steep dip angles. The Yangsan fault is clearly identified from the rivermouth of the Nakdong River to north of Yangsan City, which mostly coincides with the valley. The initial movement of the Yangsan fault is estimated to be 45 Ma ago (Eocene) based on radiometric

data (Chang et al., 1990). The Yangsan fault was activated during 42-14 Ma (Eocene-Miocene) in relation to the opening of the East Sea (Jolivet et al., 1991).




The groundwater monitoring system for earthquake monitoring has been in operation since 2010. The monitoring system comprises wells from 7.5 to 176.1 m in elevation, from 130 to 323 m in well depth, and from 6.4 to 175.2 m in depth-to-water. Also, seven monitoring wells for seawater intrusion are located 0.1 (HM1), 0.17 (SY1), 0.2 (PP1), 0.76 (SG1), 0.8 (HD1), 3.2 (JD2), and 8.5 km (SS4) off the coast (Fig. 1, Table 1). The
groundwater level is automatically monitored every minute.

Modified moving average method was used for filtering the relatively low-frequency ocean tide from the raw groundwater levels. The equation is

$$X_t^* = X_t - \frac{X_{t-n} + \cdots X_t + \cdots X_{t+n}}{2n+1} \tag{1}$$

where $X_t^*$ is the filtered groundwater level, $X_t$ is the raw groundwater level corrected for atmospheric pressure,
and $n$ is the number of samples in the moving average before and after $X_t$. In this study, changes in groundwater level caused by earthquakes were evaluated using $n=2$ in the above equation.

### 3 Results and discussion

#### 3.1 Groundwater level change caused by the Kumamoto earthquakes

An earthquake of M 7.0 occurred in Kumamoto prefecture, Kyushu, Japan, at 01:25:06, on April 16, 2016 at
local time 01:25:00 (April 15, 2016 at UTC time) (Uchide et al., 2016). Three foreshocks (M 6.2, M 5.4, and M 6.0) occurred in the Kumamoto area two days before the occurrence of the M 7.0 main earthquake. Since the main earthquake, many aftershocks have occurred, such as the M 5.4 aftershock at local time 03:03:10, on April 16, 2016 (Table 1, Fig. 1).

Depth-to-water level (DTW) data was corrected for the atmospheric pressure of the SG1 well from April 14,
2016 to April 16, 2016, and then the filtered water level (FWL) was estimated by removing tidal effects and long term tendencies using Eq. (1) (Fig. 2). The FWL reacted 2 min following the initiation of the M 7.3 Kumamoto earthquake at a distance of ~390 km from the epicenter (Fig. 3). Most changes in the groundwater level due to the Kumamoto continuous earthquake at the monitoring wells are of the oscillation type (Fig. 4). This type of groundwater level oscillation is represented by aquifer geometry, including water depth in the well,
well radius, and aquifer thickness as well as hydrogeologic properties (transmissivity and specific storage etc.). Dynamic deformation by seismic waves that compresses and expands the rock around the aquifer causes a change in pore pressure that repeatedly impacts the inflow and outflow of the well (Cooper et al., 1965; Liu et al., 1989; Brodsky et al., 2003; Manga and Wang, 2007).

According to previous studies (Montgomery and Manga, 2003; Wang and Chia, 2008), groundwater level
changes due to earthquakes are different depending on the distance from the epicenter and the magnitude. The average groundwater level change caused by the Kumamoto earthquake was 1.4 cm for the M 5.4 foreshock, 0.7 cm for the M 5.4 aftershock, 2.2 cm for M 6.0, and 3.5 cm for M 6.2, with an average water level change width of 18.6 cm caused by the M 7.0 main earthquake (Table 2). These changes in groundwater level are proportional to the magnitudes of the continuous Kumamoto earthquakes (Lee et al., 2013). The groundwater level change
caused by the M 5.4 aftershock was less than that caused by the M 5.4 foreshock because the M 5.4 aftershock

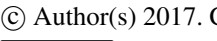



occurred only 3 h after the main earthquake (M 7.0). This is explained by the fact that other aftershocks (not observed in the wells) during the time span of the main earthquake and the M 5.4 aftershock continuously affected the aquifers and wells.

### 3.2 Groundwater level change caused by the Gyeongju M 5.4 earthquake

An M 4.9 earthquake at 19:44:32 local time (10:44:32 UTC time) on September 12, 2016 and an M 5.4 earthquake at 20:32:54 local time about 1 h after the M 4.9 earthquake occurred in Gyeongju City, South Korea (Table 1, Fig. 1). The M 5.4 earthquake was recorded as the largest earthquake since the beginning of the instrumental earthquake monitoring in 1978 in South Korea. Before the Gyeongju earthquake, the largest earthquake in Korean peninsula was the M 5.3 earthquake that occurred in the western Pyongan Book-do
province at 08:44:13, on January 8, 1980 (local time). The Korean government declared Gyeongju City as a special disaster area as a result of the M 5.4 Gyeongju earthquake and the aftershocks (more than 500) that damaged life and buildings.

Groundwater level changes caused by the M 5.4 earthquake were observed at four monitoring wells on Jeju Island. Recorded changes in groundwater level as a result of the Gyeongju earthquake were 0.7-5.4 cm (1.9 cm
at SG1, 3.6 cm at PP1, 5.4 cm at JD2, and 1.9 cm at SS4) 2-3 min after the generation of the earthquake, which then returned to the original level (Table 2, Fig. 5). Interestingly, groundwater levels of the SY1 and HD1 wells did not change during the Gyeongju earthquake, but they responded sensitively to the Kumamoto earthquake. In addition, greater water level changes caused by the M 5.4 Gyeongju earthquake were observed compared to the M 5.4 foreshock and M 5.4 aftershock of the Kumamoto earthquake. The groundwater levels at the PP, JD2, and
SS4 wells, which did not respond to the M 5.4 foreshock or the M 5.4 aftershock, responded to the Gyeongju M 5.4 earthquake showing a greater change than they exhibited in case of the Kumamoto M 6.2 earthquake, and responded similarly to how they did to the M 7.0 earthquake.

### 3.3 Behavior of the aquifers around the monitoring wells by the earthquakes

The response of water level to the Kumamoto earthquakes depended on the magnitudes of the earthquakes, as
well as on the aquifer and geological characteristics. The variations of groundwater level at the SG1, SY1, and HD1 wells, which were sensitive to the Kumamoto earthquakes, showed that the variation in groundwater level gradually increased with respect to magnitude (Fig. 6). In the case of the SY1 well, the water level change was greatly increased by the M 7.0 earthquake, indicating that the hydraulic characteristics at the well were amplified relative to the seismic energy above a critical level. This phenomenon was also observed at the HL1
well as a result of the 2011 Tohoku earthquake (M 9.0), such that the groundwater levels changed by 2.3, 9.8, and 17.7 cm as a result of the foreshocks M 6.1 and M 7.2 and the aftershock M 7.9, respectively, and the level was suddenly amplified by a M 9.0 earthquake (Fig. 7).

Based on the four geological columns at the SG1, SY1, HD1, and PP wells in the coastal area, the leaky confined aquifer model was appropriate for the study area. Consequently, transmissivity estimates were
determined by the leaky confined aquifer model and the relationship of transmissivity (T) to specific discharge (Q/s), $T = 0.99(Q/s)0.89$, proposed by Hamm et al. (2005). The hydraulic conductivity estimates were driven by T divided by the aquifer thickness using the geological columns. Using the relationship of calculated hydraulic



conductivity to groundwater level change caused by the Kumamoto M 7.0 earthquake, higher hydraulic conductivity estimates yielded higher groundwater level changes.

### 3.4 Discussion

The water level change caused by the Gyeongju earthquake was observed to be considerably larger than that caused by the Gyeongju M 5.4 earthquake and the Kumamoto M 5.4 earthquake. The southwestern extension of the Yangsan fault, in which the Gyeongju earthquake took place, extends to Jeju Island running in the NE-SW direction. Therefore, the energy of the Yangsan earthquake may have been effectively transmitted along the fault plane, resulting in a groundwater level change greater than that caused by the Kumamoto earthquake. This induced a smaller change in groundwater level resulting from the damping of the earthquake energy as the

seismic wave passed through faults with a strike in the NE direction between Korea and Japan (Fig. 1).

### 4 Conclusions

The groundwater level responses displayed different behaviors depending on the magnitude of the earthquakes and the distance from the source as well as different sensitivities according to the aquifer and geological characteristics around the monitoring wells. Particularly, for certain observations, amplification of the water

level change was observed as the magnitude (or energy) increased.
On Jeju Island, the water level change was comparatively larger as a result of the Gyeongju M 5.4 earthquake that occurred along Yangsan fault than that caused by the Kumamoto M 5.4 earthquake that occurred in the MTL fault zone. The southwestern part of the Yangsan fault that extends to Jeju Island runs in NE-SW direction, while the MTL fault zone on the Japanese islands extends to the ENE-WSW direction. Therefore, the energy of

the Yangsan earthquake may have been effectively transmitted along the fault plane, resulting in a groundwater level change that was higher than that caused by the Kumamoto earthquake, which induced a smaller change in groundwater level because of the damping of earthquake energy as the seismic wave passed through the faults between Korea and Japan.

### ACKNOWLEDGMENTS

This research was supported by the Basic Research Project (grant no. 16-3415) of the Korea Institute of Geoscience and Mineral Resources (KIGAM) and Korea Ministry of Environment as "GAIA (Geo-Advanced Innovative Action) Project (#2016000530004). The authors thank the officers of Water Resources Headquarter of Jeju Special Self-Governing Province for their help in selecting the monitoring wells and obtaining groundwater data.

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





Table 1. A list of the 2016 Kumamoto and Gyeongju earthquakes.

| Earthquake | | Magnitude (M)* | Depth (km) | Date (Local time) | Date (UTC) |
|---|---|---|---|---|---|
| Kumamoto | | 6.2 | 9.0 | 2016.04.14 21:26:35 | 2016.04.14 12:26:35 |
| | Foreshock | 5.4 | 10.0 | 2016.04.14 22:07:35 | 2016.04.14 13:07:35 |
| | | 6.0 | 8.0 | 2016.04.16 00:03:47 | 2016.04.14 15:03:47 |
| | Main earthquake | 7.0 | 10.0 | 2016.04.16 01:25:06 | 2016.04.15 16:25:06 |
| | Aftershock | 5.4 | 4.4 | 2016.04.16 03:03:10 | 2016.04.15 18:03:10 |
| Gyeongju | Foreshock | 4.9 | 13.0 | 2016.09.12 19:44:32 | 2016.09.12 10:44:32 |
| | Main earthquake | 5.4 | 13.0 | 2016.09.12 20:32:55 | 2016.09.12 11:32:55 |

UTC stands for Universal Time Coordinated
* The magnitudes were presented by USGS.

Table 2. Groundwater level changes of the observation wells on the Jeju volcanic Island as a result of the Kumamoto and Gyeongju earthquakes.

| OW | Foreshock | | | Main earthquake | Aftershock | Gyeongju |
|---|---|---|---|---|---|---|
| | M 6.2 | M 5.4 | M 6.0 | M 7.0 | M 5.4 | M5.4 |
| SG1 | 10.4 | 1.6 | 3.8 | 18.6 | 1.0 | 1.9 |
| SY1 | 5.6 | 1.1 | - | 38.2 | 0.5 | - |
| HD1 | 1.0 | - | 0.5 | 6.6 | 0.6 | - |
| PP1 | 0.5 | - | - | 4.0 | - | 3.6 |
| JD2 | 2.6 | - | - | 3.1 | - | 5.4 |
| SS4 | 0.6 | - | - | 4.2 | | 1.9 |
| Average | 3.5 | 1.4 | 2.2 | 12.5 | 0.7 | 2.7 |

OW: Observation well
HM well: data error




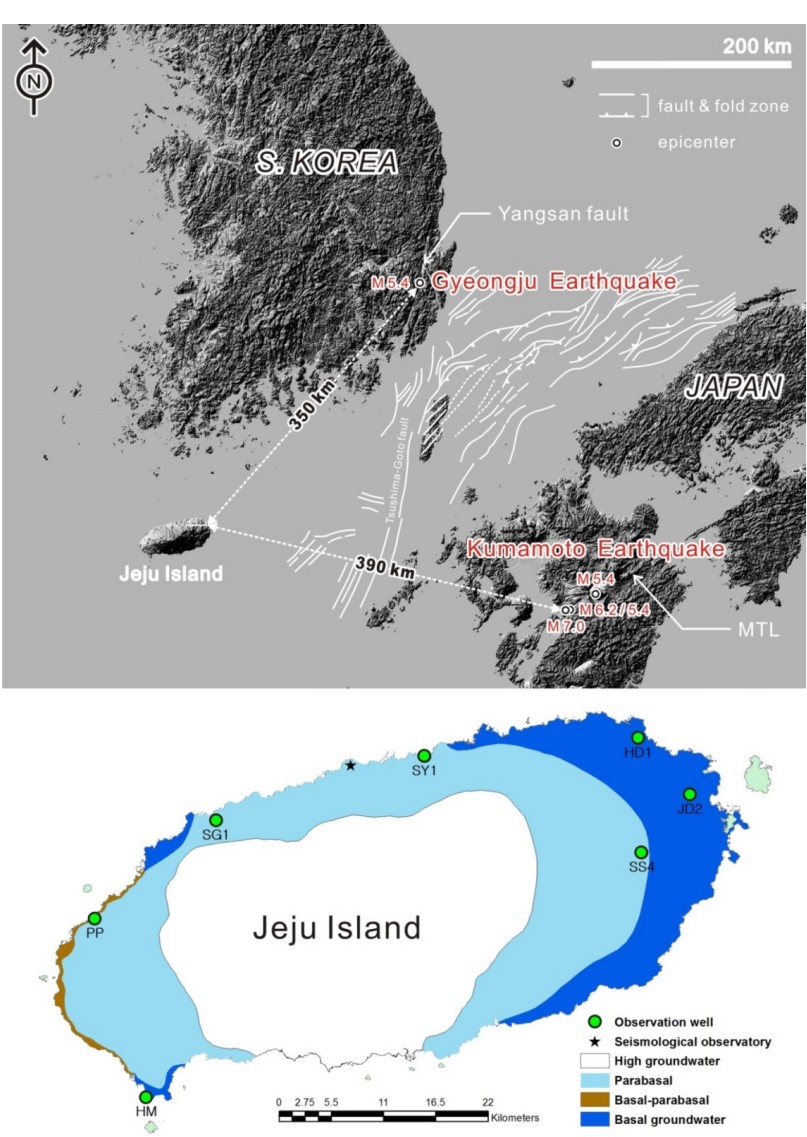

Fig 1. Map of the Kumamoto and Gyeongju earthquake epicenters with groundwater observation wells on Jeju Island.



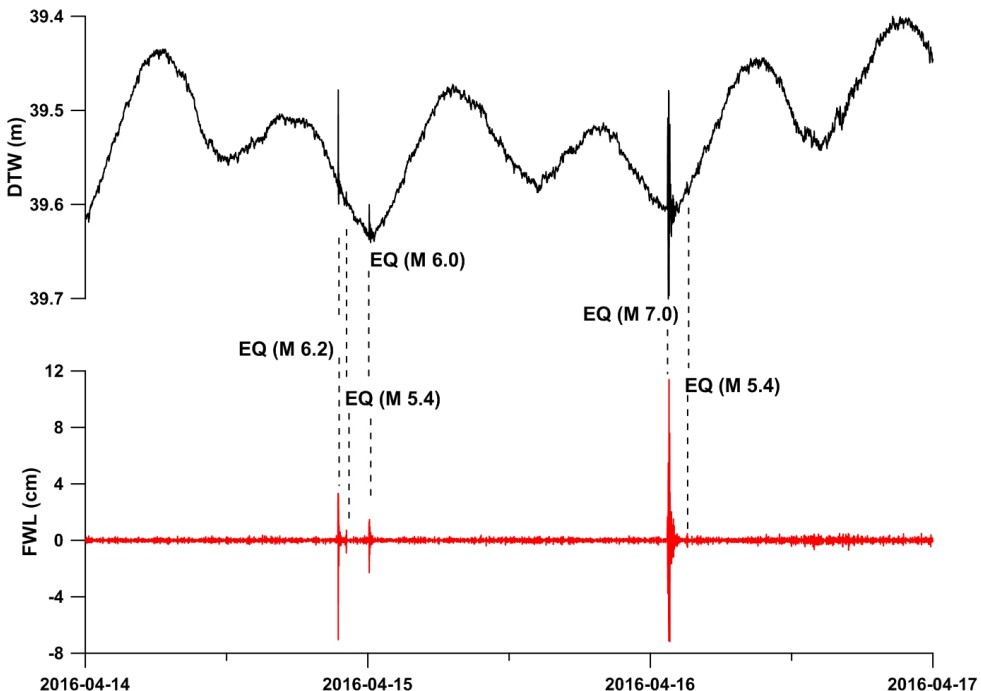

Fig 2. Changes of groundwater level related to the successive Kumamoto earthquakes at well SG1 from 14 to 16 April, 2016. The DTW (depth-to-water) data was corrected for the atmospheric pressure and the FWL (filtered water level) was estimated by removing tidal effects and long term tendencies.


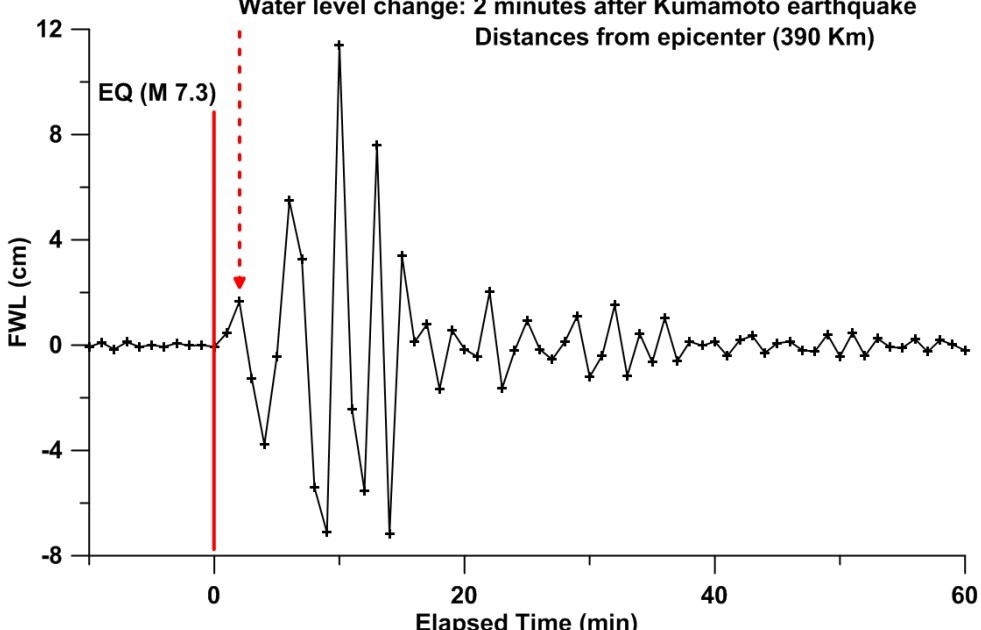

Fig 3. Changes in groundwater level at the SG1 well caused by the M 7.0 Kumamoto earthquake.

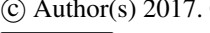


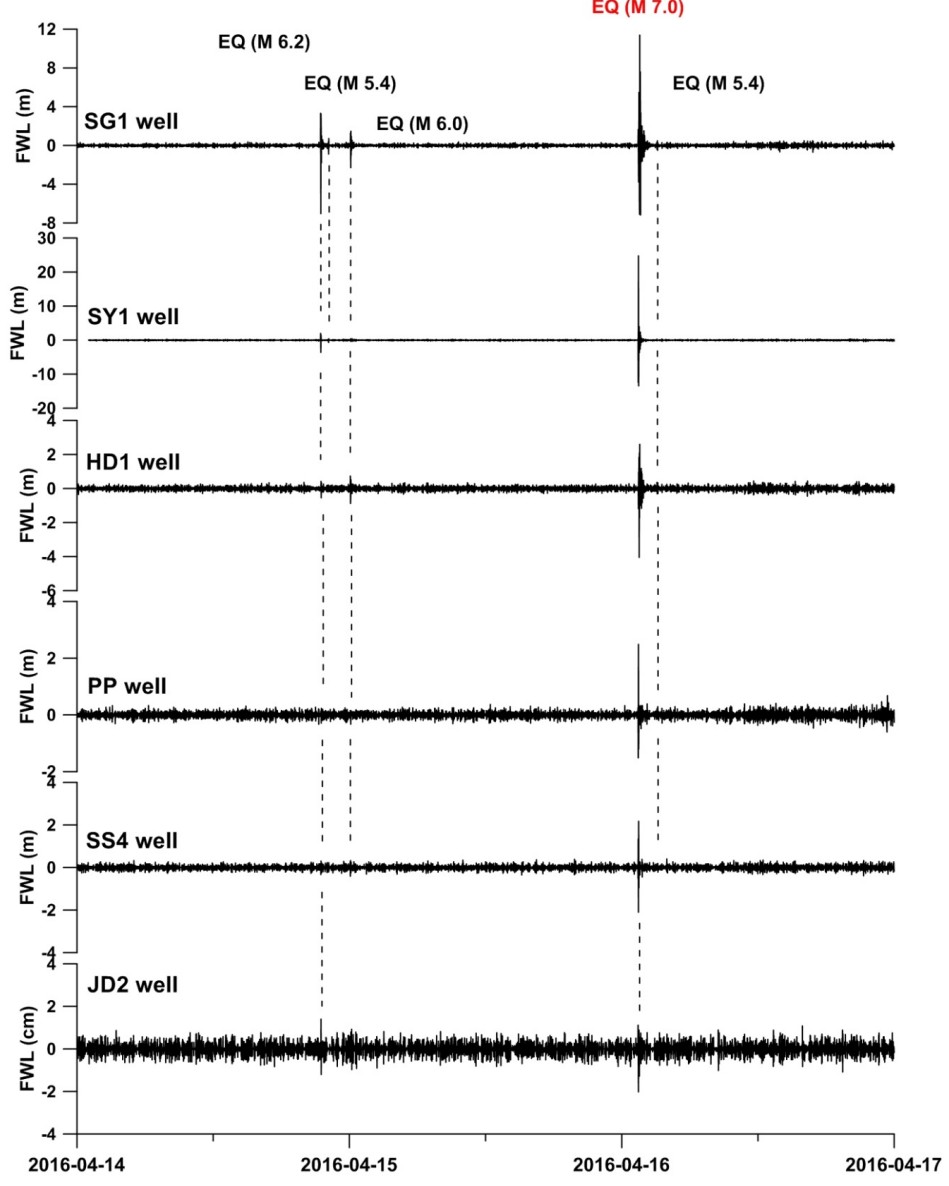

Fig 4. Changes in groundwater level related to the successive Kumamoto earthquakes at observation wells from 14 to 16 April, 2016.

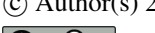


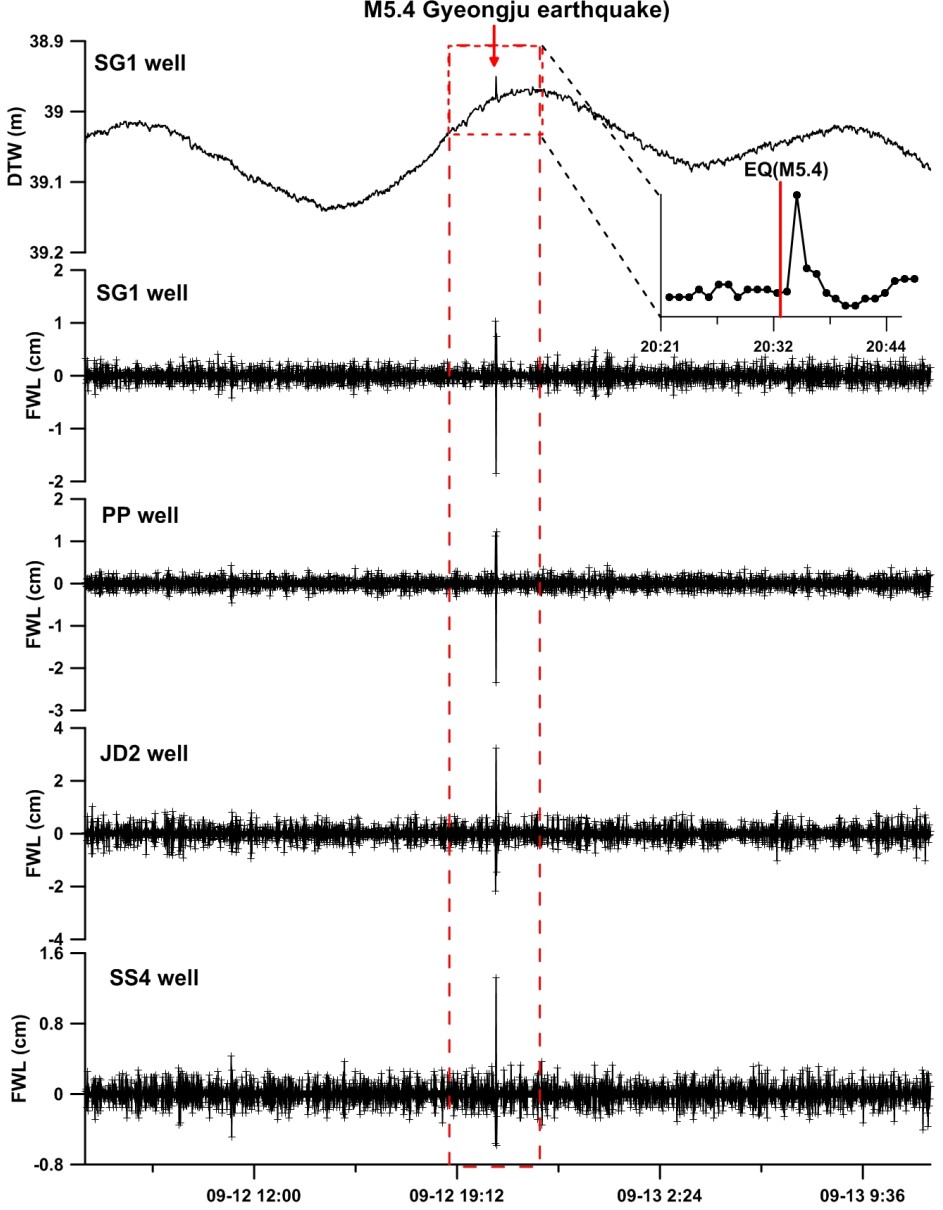

Fig 5. Groundwater level changes in the observation wells (SG1, PP, JD2, and SS4) caused by the M5.4 Gyeongju earthquake.


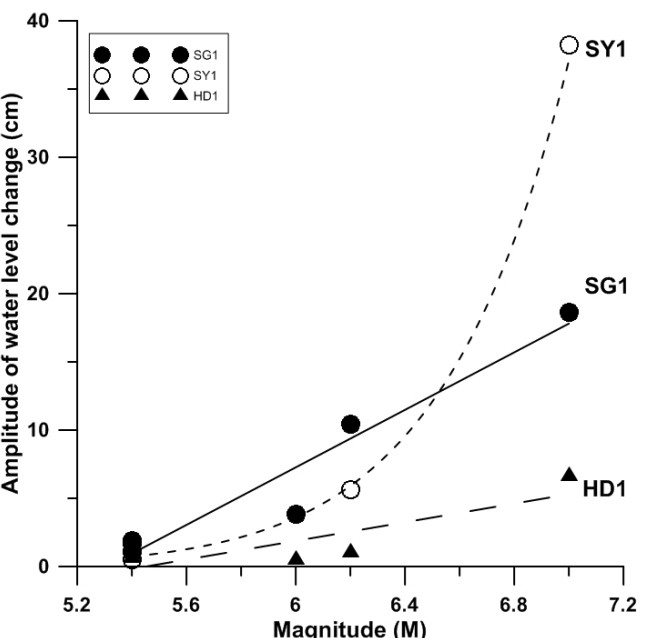

Fig 6. Groundwater level changes at the SG1, SY1, and HD1 wells vs. the Kumamoto earthquakes.

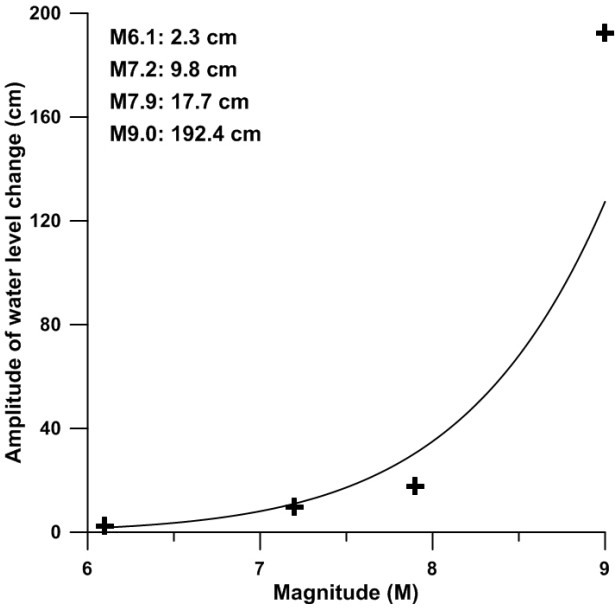

Fig. 7. Groundwater level changes at the HL1 well vs. magnitudes of the 2011 Tohoku



earthquakes.

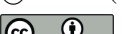



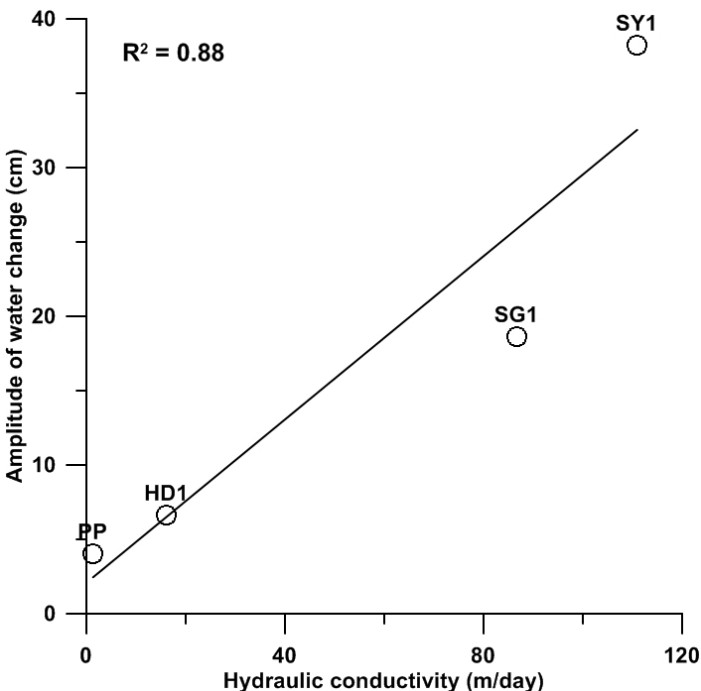

Fig. 8. Groundwater level changes vs. hydraulic conductivity estimates using leaky aquifer model.