# Peer review of "Groundwater level changes on Jeju Island associated with the Kumamoto and Gyeongju earthquakes"

_Natural Hazards and Earth System Sciences, 2017_

## Referee Comment (RC1) · Anonymous Referee #1 · 1 Mar 2017

This manuscript discussed the coseismic water level changes following several earthquakes. The authors want to explain the different response amplitudes following the same magnitude earthquakes. And they attributed to the different energy attenuation in different earthquakes. The topic of coseismic water level changes is interesting, but in my view, this manuscript did not convinced me well. Overall, more details about the hydrogeology setting of the wells and more quantitative analysis are needed in order to understand better about the mechanism of the coseismic response.

I have several comments on this manuscript, the details are as followings:

(1). The author described the lithology and groundwater wells in Jeju island, but no information is provided about aquifer and the well structure of these wells. I suggest

that a cross-section of the hydrogeology setting with location of these wells would be insightful.

(2).Equation 1, the authors use the moving average method to filter the relative low-frequency ocean tide. However, in my understanding, the ocean tide has different frequencies (ranges from low to high frequencies). What's more, I think if you want to filter the low frequency signals, why don't you use the high-pass filtering? I am doubting about the result by moving average method in filtering the low frequency data. The ocean tide can be calculated and removed by several existed programs such as Spotl, Baytap.

(3).The authors argued that the groundwater level caused by M5.4 foreshock was less than the M5.4 foreshock, and attributed this difference to the effects of M7.0 mainshock. Although the aftershock has lower amplitude than the foreshock, there are three wells have coseismic water level response following aftershock, but only two wells show response following the foreshock. The authors may need to explain this phenomenon.

(4). Page 4, "35". What's four geological columns at the SG1, SY1, HD1 and PP well, and what's the leaky confined aquifer model refer to? The authors should provide much more details about them. There are seven wells in the study, why the author only use the four well's response data to show the relation between hydraulic conductivity and water level response.

(5). The author said Xt is the raw groundwater level corrected for atmospheric pressure, but they didn't show what method was used in the correction.

(6). In Discussion section, the author argued that the larger response following Gyeongju earthquake than foreshock and after shock (M5.4) is caused by the extension of the Yangsan fault. However, from the Figure 1, I cannot see any parts of the Yangsan fault across the Jeju Island. Thus I think it is not justified to say that the energy of the Yangsan earthquake may have been effectively transmitted along the fault plane. The authors need to provide addition evidence to support their speculation. In fact, this

phenomenon is more likely caused by the hydraulic properties changes after the M7.0 earthquake. Because large earthquake will lead to the changes of aquifer properties and even the disruption of aquifer system (Brodsky, 2003; Wang et al., 2004; Elkhoury et al., 2006; Manga et al., 2012; Xue et al., 2013; Wang et al., 2016), the aquifer properties might have changed after the M7.0 earthquake, thus the different aquifer properties in the Kmmamoto M5.4 and Gyeongju M5.4 lead to the different amplitudes of water level changes. Quantitative calculation of the aquifer properties is need in order to further discuss the mechanism.

---

## Referee Comment (RC2) · Anonymous Referee #2 · 14 Mar 2017

The manuscript presents observation data of groundwater level changes by the Kumamoto and Gyeongju esrthquakes. And authors proposed that the differences in groundwater level change are mainly due to the fault strutures of MTL and Yangsan fault, respectively, presented in Figure 1. - This interpretation sounds perceptable, but needs more data to support the argument.

(P.5, line 1∼2) Authors stated that "higher hydraulic conductivity estimates yielded higher groundwater level changes". - Higher hydraulic conductivities of aquifer matrix mean higher potentials of seismic energy to be reduced by groundwater along its pass ways. Then, how can the higher K cause higher water-level changes?

Miscellaneous: - In Table 2, data need proper units. - In addition, in the groundwater

level monitoring, the sensitity should be given here to support less than 1 cm of change being significant.

[Conclusion] Groundwtaer level change is affected by numerous factors including tectonic settings, local geology of monitoring sites, monitoring well design and hydrogeologic properties of monitoring intervals. Thus, to make the argument of structural difference as the main cause of the difference of water-level changes, this manucript should present some quantitative evidences. Otherwise, it could be just an observation report.

---

## Author Comment (AC1) · 12 May 2017

Dear Reviewer 1,

Thank you for reviewing our manuscript and for providing useful comments. Below, we have outlined the replies to your comments. Also, we revised our manuscript based on your comments.

(1) The authors described the lithology and groundwater wells in Jeju Island, but no information is provided about aquifer and the well structure of these wells. I suggest that a cross-section of the hydrogeology setting with location of these wells would be insightful.

1) Reply - We agree with your comments. Accordingly, we added Fig. 2 (Geologic logs of the monitoring wells). As described in the revised manuscript (the lines 4 - 9 on the page 3), based on the geological logs of the monitoring wells that are mostly drilled to UF (Unconsolidated Formation), porous lava flows of repeatedly accumulated layers cover the basement rocks. Especially, in the higher altitude area, more abundant lava flows appear with the existence of many volcanic units (including clinker, scoria, sediments, pyroclastites, and hyaloclasitetes etc.) between the lava flows. Consequently, groundwater moves along complicated flow paths under the influence of numerous hydraulic parameters. (See the figure below)

(2) Equation 1, the authors use the moving average method to filter the relative low frequency ocean tide. However, in my understanding, the ocean tide has different frequencies (ranges from low to high frequencies). What's more, I think if you want to filter the low frequency signals, why don't you use the high-pass filtering? I am doubting about the result by moving average method in filtering the low frequency data. The ocean tide can be calculated and removed by several existed programs such as Spotl, Baytap.

2) Reply - We agree with your comments. Tidal effect was eliminated from the measured groundwater level time series, by considering ocean tidal prediction and using T_TIDE MATLAB code (Pawlowicz et al., 2002) that was available from the server (http://www.iamg.org/CG-Editor/index.htm or http://www.ocgy.ubc.ca/~rich). (See the figure below).

(3) The authors argued that the groundwater level caused by M 5.4 foreshock was less than the M 5.4 aftershock, and attributed this difference to the effects of M 7.0 main shock. Although the aftershock has lower amplitude than the foreshock, there are three wells have coseismic water level response following aftershock, but only two wells show response following the foreshock. The authors may need to explain this phenomenon.

3) Reply - Average groundwater level changes of 1.4 cm with response at two wells

for the M 5.4 foreshock and 0.7 cm with three wells for the M 5.4 aftershock can be explained by the difference of focuses of the M5.4 foreshock (10.0 km) and M5.4 aftershock (4.4 km) because the deeper is the focus, the farther is transmitted the seismic energy.

(4) Page 4, "35". What's four geological columns at the SG1, SY1, HD1 and PP well, and what's the leaky confined aquifer model refer to? The authors should provide much more details about them. There are seven wells in the study, why the author only use the four well's response data to show the relation between hydraulic conductivity and water level response.

4) Reply - Based on the geological columns of the seven monitoring wells (Fig. 2) and the groundwater level change vs. hydraulic conductivity (Fig. 9), we identified a positive proportionality between groundwater level change and hydraulic conductivity. In addition, the wells of greater groundwater level change displayed oscillation type with higher transmissivity.

(5) The author said Xt is the raw groundwater level corrected for atmospheric pressure, but they didn't show what method was used in the correction.

5) Reply - The raw groundwater level was corrected for barometric pressure effect, by simultaneous atmospheric pressure measurement with the groundwater level measurement. Please refer Fig. 3. (See the figure below).

Fig 3. Changes of groundwater level related to the successive Kumamoto earthquakes at well SG1 from 7 to 24 April, 2016. The DTW (depth-to-water) means the corrected groundwater level for the atmospheric pressure. The residual groundwater level series mean the result by having removed the tidal effect. The FWL (filtered water level) means the level having removed long term tendency by the modified moving average method.

(6) In Discussion section, the author argued that the larger response following

Gyeongju earthquake than foreshock and after shock (M5.4) is caused by the extension of the Yangsan fault. However, from the Figure 1, I cannot see any parts of the Yangsan fault across the Jeju Island. Thus I think it is not justified to say that the energy of the Yangsan earthquake may have been effectively transmitted along the fault plane. The authors need to provide addition evidence to support their speculation. In fact, this phenomenon is more likely caused by the hydraulic properties changes after the M7.0 earthquake. Because large earthquake will lead to the changes of aquifer properties and even the disruption of aquifer system (Brodsky, 2003; Wang et al., 2004; Elkhoury et al., 2006; Manga et al., 2012; Xue et al., 2013; Wang et al., 2016), the aquifer properties might have changed after the M7.0 earthquake, thus the different aquifer properties in the Kumamoto M5.4 and Gyeongju M5.4 lead to the different amplitudes of water level changes. Quantitative calculation of the aquifer properties is need in order to further discuss the mechanism.

6) Reply - Based on tectonics, several fault fold zones including Tsushima-Goto fault of NNE-SSW direction lies between Jeju Island and Kumamoto and greatly attenuated the energy by the Kumamoto M 5.4 earthquake. By contrast, the energy by the Gyeongju earthquake was effectively transmitted due to the parallel extended Yangsan fault west Tsushima-Goto fault that elongates to the east of Jeju Island (Kim et al., 2016). We also agree with you about that the possibility of the different aquifer responses may affected the different water level changes by the Kmmamoto M5.4 and Gyeongju M5.4. Nevertheless, it is thought that the tectonics between the Korean peninsula, Jeju Island, and Kumamoto is a major factor of the groundwater level change by the Kumamoto and Gyeongju earthquakes. (Refer the lines 18-23, page 5.)

Reference

Pawlowicz, R., Beardsley, B., and Lentz, S.: Classical tidal harmonic analysis including error estimates in MATLAB using T_TIDE, Computers & Geosciences, 28, 929-937, 2002.

Please also note the supplement to this comment:
http://www.nat-hazards-earth-syst-sci-discuss.net/nhess-2017-28/nhess-2017-28-AC1-supplement.pdf

———————————————————

Interactive
comment

[Figure]

SS4

150.0

100.0

50.0

Sea
level
0.0

HM1    SY1    PP1    HD1    SG1    JD2

-50.0

-100.0

-150.0

**Stratigraphy**

- Soil
- Basalts
- Scoria
- Clinker
- Pyroclastite
- Hyaloclastites
- Sand
- Mud
- Sandstone
- Tuff
- SGF
- UF

**Fig. 1.** RC1_reply_(1) question (Fig. 2)

**SG1**

Kumamoto earthquake

Original time series
Tidal prediction
Residual series

**Fig. 2.** RC1_reply_(2) question

[Figure]

[Figure]

**Fig. 3.** RC1_reply_(3) question (Fig. 5)

---

## Author Comment (AC2) · 12 May 2017

Dear Reviewer 2,

Thank you for reviewing our manuscript and for providing useful comments. Below, we have outlined the replies to your comments. Also, we revised our manuscript based on your comments.

(1) The manuscript presents observation data of groundwater level changes by the Kumamoto and Gyeongju earthquakes. And authors proposed that the differences in groundwater level change are mainly due to the fault structures of MTL and Yangsan fault, respectively, presented in Figure 1. – this interpretation sounds perceptible, but

needs more data to support the arguments.

1) Reply We agree with your comments. We added or modified manuscript. Based on tectonics, several fault fold zones including Tsushima-Goto fault of NNE-SSW direction lies between Jeju Island and Kumamoto and greatly attenuated the energy by the Kumamoto M 5.4 earthquake. By contrast, the energy by the Gyeongju earthquake was effectively transmitted due to the parallel extended Yangsan fault west Tsushima-Goto fault that elongates to the east of Jeju Island (Kim et al., 2016). We also agree with you about that the possibility of the different aquifer responses may affected the different water level changes by the Kmmamoto M5.4 and Gyeongju M5.4. Nevertheless, it is thought that the tectonics between the Korean peninsula, Jeju Island, and Kumamoto is a major factor of the groundwater level change by the Kumamoto and Gyeongju earthquakes. The above description is provided on the lines 18-23 of page 5 in the revised version.

(2) (p.5, line 1∼2) Authors stated that "higher hydraulic conductivity estimates yielded higher groundwater level changes". – higher hydraulic conductivities of aquifer matrix mean higher potentials of seismic energy to be reduced by groundwater along its pass ways. Then, how can the higher K cause higher water-level changes?

2) Reply We added or modified manuscript. Based on the geological columns of the seven monitoring wells (Fig. 2) and the groundwater level change vs. hydraulic conductivity (Fig. 9), we identified a positive proportionality between groundwater level change and hydraulic conductivity. When aquifer has a sufficiently high transmissivity, groundwater may easily flow into and out of the well, causing resonant motions in the hydraulic head (Cooper et al., 1965; Liu et al., 1989). Various transmissivity values may account for the different oscillation characteristics of the wells (Wang et al., 2009; Lee et al., 2013). In general, wells of higher transmissivity show more sensitive response to earthquakes even though the correlation is not great (Wang and Manga, 2009).

(3) Miscellaneous: - In Table 2, data need proper units. – In addition, in the groundwater level monitoring, the sensitivity should be given here to support less than 1cm of change being significant.

3) Reply According to your comments, we indicated the unit, cm, in Table 2. The specifications of the observation equipment are as follows: pressure range 10m, Accuracy $\pm$1.0cm, Resolution 0.2cm, Temperature range: -20$\sim$ +80°C, $\pm$1.0°C, 0.01°C (line 5-6 of page 3).

(4) [Conclusion] Groundwater level change is affected by numerous factors including tectonic settings, local geology of monitoring sites, monitoring well design and hydrogeologic properties of monitoring intervals. Thus, to make the argument of structural difference as the main cause of the difference of water-level changes, this manuscript should present some quantitative evidences. Otherwise, it could be just an observation report.

4) Reply We agree with your comments. We added or modified manuscript. Groundwater level change is affected not only tectonic settings but also local geology of monitoring sites, monitoring well design and hydrogeologic properties of monitoring intervals. Yet, in our study, four wells (SG1, PP1, JD2, and SS4) except two wells (SY1 and HD1) showed greater groundwater level changes to the Gyeongju M 5.4 earthquake than the foreshock and aftershock M 5.4 of the Kumamoto earthquake. This indicates that in the monitoring wells, tectonic setting is more dominant factor of groundwater level change to the Gyeongju and Kumamoto earthquakes, even if there exist local influences such as site geology around monitoring wells, monitoring well design, and hydrogeologic properties of the monitoring intervals. We revised our manuscript such description on the lines 11-14 and 33-35 of page 5.

Please also note the supplement to this comment:
http://www.nat-hazards-earth-syst-sci-discuss.net/nhess-2017-28/nhess-2017-28-AC2-supplement.pdf